# On the Riemann Function

**Peter J. Zeitsch**

School of Mathematics and Statistics, University of Sydney, Sydney, NSW 2006, Australia;
peterz@maths.usyd.edu.au

**Abstract:** Riemann's method is one of the definitive ways of solving Cauchy's problem for a second order linear hyperbolic partial differential equation in two variables. The first review of Riemann's method was published by E.T. Copson in 1958. This study extends that work. Firstly, three solution methods were overlooked in Copson's original paper. Secondly, several new approaches for finding Riemann functions have been developed since 1958. Those techniques are included here and placed in the context of Copson's original study. There are also numerous equivalences between Riemann functions that have not previously been identified in the literature. Those links are clarified here by showing that many known Riemann functions are often equivalent due to the governing equation admitting a symmetry algebra isomorphic to $SL(2,R)$. Alternatively, the equation admits a Lie-Bäcklund symmetry algebra. Combining the results from several methods, a new class of Riemann functions is then derived which admits no symmetries whatsoever.

**Keywords:** Riemann function; Riemann's method; hypergeometric function of several variables; point symmetry; generalized symmetry

## 1. Introduction

The interest in Riemann's method is long-standing. The reason is that once the Riemann function is determined, the governing equation can be solved for Cauchy data on any other non-characteristic curve. The value of such a property means that Riemann's method continues to draw the attention of investigators today. Some applications include solving electromagnetic problems exhibiting rotational symmetry [1], finding existence criteria for the eigenvalues of the solution of focal point problems [2], solving for the solution of transient plane waves [3], and the inverse problem of scattering theory [4–12]. More recently, Riemann's method has been applied to the solution of coupled Korteweg-de Vries equations [13], to boundary value problems for the non-homogeneous wave equation [14–18], to the solution of the non-linear Schrödinger equation [19,20], and modelling hyperbolic quasi-linear equations [21–23].

Copson [24] wrote the first review of Riemann's method in 1958. In total, he listed six different techniques to solve for the Riemann function. However, three other approaches were missed by Copson. They are included here for completeness. Since 1958, another four methodologies have emerged for finding Riemann functions. Often the resulting Riemann functions are not new but rather a reduction of some more general Riemann function or obtainable from another Riemann function by a change of variables. This fact is frequently overlooked in the literature. Here, the new methods are outlined to extend Copson's review and the equivalences are clarified. One finding is that the Riemann functions are equivalent because the governing equation admits a symmetry algebra isomorphic to $SL(2,R)$. Alternatively, the equation admits a Lie-Bäcklund symmetry algebra. By combining several of the solution techniques, a new class of Riemann functions is then obtained that admits no symmetries whatsoever.

Consider the partial differential equation (PDE) in characteristic variables

$$L[U] = U_{rs} + a(r,s)U_r + b(r,s)U_s + c(r,s)U = 0. \tag{1}$$

The aim is to represent a solution $U$ at a point $P_0$ in terms of the initial data, which are the values of $U$ and one outgoing derivative of $U$ on the initial curve $C$. Thus, both $U_r = c_1$ and $U_s = c_2$, where $c_1$ and $c_2$ are constants, are known on $C$ from such data as shown in Figure 1. Take the adjoint

$$L^*[V] = V_{rs} - (aV)_r - (bV)_s + cV, \tag{2}$$

and define $V = R(r,s,r_0,s_0)$ such that

$$L^*[R] = 0, \tag{3}$$

$$\frac{\partial}{\partial r}R(r,s,r_0,s_0) = b(r,s_0)R(r,s,r_0,s_0), \quad \text{on} \quad s = s_0, \tag{4}$$

$$\frac{\partial}{\partial s}R(r,s,r_0,s_0) = a(r_0,s)R(r,s,r_0,s_0), \quad \text{on} \quad r = r_0, \tag{5}$$

$$R(r_0,s_0,r_0,s_0) = 1. \tag{6}$$

$R(r,s,r_0,s_0)$ is called the Riemann function. It is the solution of the characteristic boundary value problem for the adjoint equation denoted by (3)–(6). $R(r,s,r_0,s_0)$ will hold for arbitrary initial values given along an arbitrary noncharacteristic curve C. The Cauchy problem then has a unique solution given by

$$U(r_o,s_0) = \frac{1}{2}\left([UR]_A + [UR]_B\right) + \frac{1}{2}\int_{AB}\left(RU_s - UR_s\right)dr + \left(RU_r - UR_r\right)ds. \tag{7}$$

An equivalent formulation to that just given can also be obtained by using the canonical variables

$$r = y + x, \qquad s = y - x. \tag{8}$$

With this change of variables, (1) becomes

$$U_{yy} - U_{xx} + a'(x,y)U_y - b'(x,y)U_x + c'(x,y)U = 0, \tag{9}$$

where $a' = 2(a+b)$, $b' = 2(b-a)$, $c' = 4c$. The Riemann function, $R(x,y,x_0,y_0)$, must now satisfy

$$L^*[R] = R_{yy} - R_{xx} - (a'R)_y + (b'R)_x + c'R = 0, \tag{10}$$

$$R_x + R_y = \frac{1}{2}(a' + b')R \quad \text{on} \quad y - y_0 = x - x_0, \tag{11}$$

$$R_x - R_y = -\frac{1}{2}(a' - b')R \quad \text{on} \quad y - y_0 = -(x - x_0), \tag{12}$$

$$R(x_0,y_0,x_0,y_0) = 1. \tag{13}$$

In the literature, both the characteristic (1) and the canonical form (9) are used.

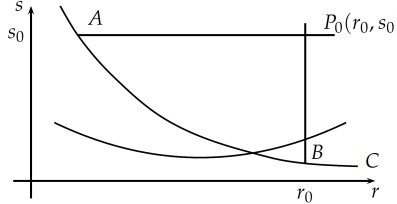

**Figure 1.** The Initial Curve.

## 2. Copson's Review

Copson [24] published the first review of Riemann's method in 1958. In all, Copson listed six different methods for finding the Riemann function. They were:

1. Riemann's original method [25], which was based on the fact that the Riemann function does not depend in any way on the curve carrying the Cauchy data. Solving the Cauchy problem by some other means for a special curve (e.g., a straight line) then yields the Riemann function by a comparison of the two solutions. Riemann only gave explicit formulae for the Riemann function in the two cases which interested him from gas dynamics. The most famous of these is the Euler-Poisson-Darboux equation (EPD)

$$U_{yy} - U_{xx} - \frac{m(1-m)}{x^2} U = 0, \tag{14}$$

   which has Riemann function

$$R(x, y, x_0, y_0) = {}_2F_1(m, 1-m; 1; z_0), \tag{15}$$

   where

$$z_0 = \frac{(y - y_0)^2 - (x - x_0)^2}{4xx_0} \tag{16}$$

   and ${}_2F_1$ is the hypergeometric function (see the Appendix A for details of all the hypergeometric functions used in this article). Equation (15) was derived via a Fourier cosine transform.

2. Hadamard [26] showed that the coefficient of the logarithmic term in his elementary solution is the Riemann function for the adjoint equation.

3. For separable equations, Copson found that it is straightforward to construct an integral equation whose unique solution is the Riemann function.

4. Chaundy [27–32] was able to construct the Riemann function for a number of equations by the use of symbolic operators and power series. Of particular note was

$$U_{rs} + \left[ \frac{m_1(1-m_1)}{(r+s)^2} - \frac{m_2(1-m_2)}{(r-s)^2} + \frac{m_3(1-m_3)}{(1-rs)^2} - \frac{m_4(1-m_4)}{(1+rs)^2} \right] U = 0, \tag{17}$$

   for which the Riemann function is given by

$$R(r, s, r_0, s_0) = F_B(m_1, m_2, m_3, m_4, 1-m_1, 1-m_2, 1-m_3, 1-m_4, 1, z_1, z_2, z_3, z_4) \tag{18}$$

   where $F_B$ is a Lauricella hypergeometric function of four variables [33] and

$$z_1 = -\frac{(r-r_0)(s-s_0)}{(r+s)(r_0+s_0)}, \qquad z_2 = \frac{(r-r_0)(s-s_0)}{(r-s)(r_0-s_0)}, \tag{19}$$

$$z_3 = -\frac{(r-r_0)(s-s_0)}{(1-rs)(1-r_0s_0)}, \qquad z_4 = \frac{(r-r_0)(s-s_0)}{(1+rs)(1+r_0s_0)}. \tag{20}$$

5. Mackie [24] constructed complex integral solutions of certain equations. An appropriate choice of contour results in the Riemann function.

6. Titchmarsh [34] gave a direct solution for the Riemann function of the equation of damped waves by means of a complex Fourier integral.

Equations (14) and (17) have been singled out here as they occur repeatedly in the literature. Equation (14) is a subcase of (17) in canonical variables, where $m_2 = m_3 = m_4 = 0$. In fact in 1958, (17) was the most general self-adjoint equation for which the Riemann function was known.

## 2.1. Self Adjoint Riemann Functions

In 1979, a summary of all the known self-adjoint Riemann functions was published by Lanckau [35]. For an arbitrary equation,

$$U_{rs} + C(r,s)U = 0, \tag{21}$$

the cases he listed were:

1. $C(r,s) = C_0$, which has Riemann function

$$R(r,s,r_0,s_0) = J_0\left(2\sqrt{z_5}\right), \tag{22}$$

   where $C_o$ is an arbitrary constant, $J_0$ is the modified Bessel function of zero order and

$$z_5 = C_0(r - r_0)(s - s_0). \tag{23}$$

2. $C(r,s) = m(1-m)(r+s)^{-2}$, which has Riemann function

$$R(r,s,r_0,s_0) = P_{-m}\left(1 - 2z_1\right), \tag{24}$$

   and $P_{-m}$ is the Legendre function of order $-m$, with $z_1$ given by (19). This characteristic form is also equivalent to the canonical representation (14) and (15) of the EPD equation.

3. $C(r,s) = -C_0 + m(1-m)(r+s)^{-2}$, which has Riemann function,

$$R(r,s,r_0,s_0) = \Xi_2\left(m, 1-m, 1, z_5, z_1\right), \tag{25}$$

   where $\Xi_2$ is a hypergeometric function of two variables [36] and $z_5$ and $z_1$ are given by (23) and (19) respectively.

4. $C(r,s) = m(1-m)(r+s)^{-2} - n(1-n)(r-s)^{-2}$, which has Riemann function

$$R(r,s,r_0,s_0) = F_3\left(m, n, 1-m, 1-n, 1, z_1, z_2\right), \tag{26}$$

   and $F_3$ is a hypergeometric function of two variables [36], with $z_1$, $z_2$ given by (19). The representation of its Riemann function in terms of $F_3$ was first obtained by Henrici [37].

5. $C(r,s) = C_0 + m(1-m)(r+s)^{-2} - n(1-n)(r-s)^{-2}$, which has Riemann function

$$R(r,s,r_0,s_0) = F_B\left(m, n, 1-m, 1-n, 1, z_5, z_1, z_2\right), \tag{27}$$

   where $F_B$ is a Lauricella hypergeometric function [36] of three variables $z_5$, $z_1$ and $z_2$, which are given by (23) and (19).

Each of the above cases is easily obtained from Chaundy's equation (17). Consider case four. If $m_3 = m_4 = 0$ in (17), the corresponding terms in the multiple power series (18) are replaced by unity. Consequently, (17) reduces automatically to the equation given in case four and (18) becomes (26). This is likewise for case two. Cases one and three require a confluence. Starting with an equation in the form of case two, make the change of variables

$$r - s = \frac{\epsilon^2}{c_0} + (r' - s').$$

Taking the limit as $\epsilon \to \infty$ yields the equation from case one, with the corresponding Riemann function (22). A similar argument shows that case three is obtainable from case four. For case five, start with Equation (17) and set $m_4 = 0$. Make the change of variables

$$
\begin{aligned}
r &= \epsilon r' \\
s &= \epsilon s' \\
m_3(1 - m_3) &= \epsilon^{-2} C_0.
\end{aligned}
$$

and let $\epsilon \to 0$ to obtain the equation from case five.

The one equation listed by Lanckau, which has not previously been linked to Chaundy's equation and its contractions, is

$$
C(r, s) = \sum_{k=1}^{4} q_k(1 + q_k) f_k^{-2}(r, s),
$$

with

$$
f_k(r, s) = a_k rs + b_k r + c_k s + d_k,
$$

where $q_k, a_k, b_k, c_k, d_k$ are constants subject to the conditions that

$$
a_k d_k - b_k c_k = 1, \qquad \text{for all } k, \tag{28}
$$

and

$$
a_l d_k - b_l c_k + a_k d_l - b_k c_l = 0, \qquad \text{for all } k \text{ and } l \neq k. \tag{29}
$$

This equation was originally published by Püngel [38,39].

Neither Püngel nor Lanckau made any connection from this case to Chaundy's Equation (17). However, standard calculations show that Püngel's equation is Möbius equivalent to Chaundy via

$$
\bar{r} = \frac{\alpha_1 r + \beta_1}{\gamma_1 r + \delta_1}, \qquad \bar{s} = \frac{\alpha_2 s + \beta_2}{\gamma_2 s + \delta_2}.
$$

Equation (28) acts as a normalization constraint while (29) restricts the parameters to Chaundy. Püngel's case is indicative of the historic difficulty of demonstrating equivalences between Riemann functions.

## 3. Methods not Included by Copson

The six approaches listed by Copson [24] were all that were known to him. Likewise, the equations listed in Section 2.1 were the only cases he was aware of, apart from trivial changes of the dependent or independent variables. However, another three methods were not included by Copson. In 1937, Courant and Hilbert [40] derived the Riemann function for the Telegrapher's equation via Lie-Point symmetries; although this was only really made clear in a 1962 translation of their benchmark text "Methoden Der Mathematischen Physik II". Their approach represents the first use of symmetry groups for Riemann's method. Cohn [41] developed an iterative technique to obtain the Riemann function. Olevskiǐ [42] also derived an addition theorem for the Riemann function of an equation based on the Riemann functions of two simpler separable equations.

### 3.1. The Telegrapher's Equation

In the original German text, Courant and Hilbert referred to their method as "der Symmetrie der Differentialgleichung den ansatz". But what this in fact means is a little obscure. In 1937, the process

of similarity reduction, instigated by Lie [43], was still embryonic. To understand what was intended, the expanded 1962 translation is more instructive. Take the Telegrapher's equation

$$L[U] = U_{rs} + C_0 U = 0, \tag{30}$$

with constant $C_0$. Courant and Hilbert's argument is then the following. Since the operator $L$ has constant coefficients, $R(r, s, r_0, s_0)$ depends only on the relative position of the points $(r, s)$ and $(r_0, s_0)$. Moreover, letting $(r_0, s_0)$ be the origin, we observe that if

$$V(r, s) = R(r, s, 0, 0)$$

satisfies conditions (3) to (6) imposed on the Riemann function, then

$$W(r, s) = V(\alpha r, \alpha^{-1} s)$$

also satisfies them. Clearly $L^*[V(r, s)] = 0$ implies $L^*[W] = 0$ so that (3) is satisfied. Condition (4) means (since $a = b = 0$) that $V = R$ remains constant along the coordinate axes. If $V$ has this property, then $W$ also has it; and finally $V(0, 0) = 1$ implies $W(0, 0) = 1$. Since these conditions determine Riemann's function uniquely, it follows that $W(r, s) = V(r, s)$ is a function of $rs$. For general $(r_0, s_0)$, the Riemann function then has the form

$$R(r, s, r_0, s_0) = f(z),$$

where

$$z = (r - r_0)(s - s_0).$$

The equation $L^*[R] = 0$ then yields the equation $zf'' + f' + C_0 f = 0$ for $f$ which, if we set $\lambda = \sqrt{4C_0 z}$, becomes Bessel's equation

$$\frac{d^2 f}{d\lambda^2} + \frac{1}{\lambda} \frac{df}{d\lambda} + f = 0,$$

which has solution

$$f = J_0(\lambda).$$

Consequently the Riemann function is

$$V(r, s, r_0, s_0) = J_0 \left( \sqrt{4C_0 (r - r_0)(s - s_0)} \right).$$

Clearly what Courant and Hilbert had in mind was that the PDE for the Riemann function and the associated boundary conditions are invariant under a three-parameter Lie transformation group. The details of their argument reflect the emerging nature of symmetry groups as a method of solution in this area. More systematic use of symmetry groups to find Riemann functions have subsequently been developed by Bluman [44], Daggit [45], Ibragimov [46], and Iwasaki [47], which will all be explored in Section 4.

### 3.2. Successive Iterations and the Banach Fixed Point Principle

In 1946, Cohn [41] considered equations of the form

$$U_{rs} + H(r + s)U = 0, \tag{31}$$

where $H$ is an arbitrary function of its argument to be determined. Cohn sought to prove the existence of $R(r, s, r_0, s_0)$ by iteration. Hence, he rewrote (31) and condition (6) as the single equation

$$R(r, s, r_0, s_0) = 1 - \int \int_{B[r,s]} R(r_1, s_1, r_0, s_0) H(r_1 + s_1) dr_1 ds_1, \tag{32}$$

where $B[r, s]$ is the rectangle formed by the horizontal and vertical lines through $(r, s)$ and $(r_0, s_0)$. By continual recursive substitution of the right hand side of (32) into itself, he obtained the formal iterated series

$$R(r, s, r_0, s_0) = 1 - \int \int_{B[r,s]} H(r_1 + s_1) dr_1 ds_1$$
$$+ \int \int_{B[r,s]} H(r_1 + s_1) dr_1 ds_1 \int \int_{B[r_1, s_1]} H(r_2 + s_2) dr_2 ds_2 - \ldots \tag{33}$$

The series (33) for $R$ begins with the terms $1 - \Delta + \ldots$ where

$$\Delta = \int \int_{B[r,s]} H(r_1 + s_1) dr_1 ds_1.$$

This iterative process is also known as the Banach fixed point principle [48]. Now $\Delta$ equals zero, whenever $r = r_0$ or $s = s_0$, thus $R(r, s, r_0, s_0)$ is also a constant for these values of $r$ or $s$. This led Cohn to try a Riemann function of the form

$$R(r, s, r_0, s_0) = R(\Delta, r_0, s_0). \tag{34}$$

Substituting (34) into (31) and (6), one finds that

$$\frac{\Delta_r \Delta_s}{H} \frac{d^2 R}{d\Delta^2} + \frac{dR}{d\Delta} + R = 0,$$
$$R(0, r_0, s_0) = 1.$$

Expanding $H(r + s) = H(\omega)$ as a power series in $\omega$, the coefficients of the series must vanish, yielding the non-linear ordinary differential equation for $H$

$$\frac{d}{d\omega} \left[ \frac{1}{H} \frac{d^2 H}{d\omega^2} - \frac{3}{2H^2} \left( \frac{dH}{d\omega} \right)^2 \right] = 0,$$

which has solution

$$H(\omega) = \frac{-\lambda(\lambda + 1)\mu^2}{\sinh^2 \mu(\omega + \nu)}, \tag{35}$$

and $\lambda, \mu, \nu$ are real valued constants. After a few further calculations, Cohn found that the Riemann function for (31), where $H$ is given by (35), is

$$R(r, s, r_0, s_0) = {}_2F_1(-\lambda, 1 + \lambda; 1; \Phi), \tag{36}$$

and

$$\Phi = -\frac{\sinh \mu(r - r_0) \sinh \mu(s - s_0)}{\sinh \mu(r + s + \nu) \sinh \mu(r_0 + s_0 + \nu)}. \tag{37}$$

The form of $H$ immediately suggests a few simpler limiting cases. For example, take $\lambda \to \lambda e^{\mu \nu}$ and rename $\lambda \to \sqrt{\lambda}/\mu$, $\mu \to -\mu/2$, then if $\nu \to \infty$ we obtain $H(r+s) = -\lambda e^{\mu(r+s)}$. Applying the limiting process of confluence, the Riemann function is then

$$R(r,s,r_0,s_0) = J_0 \left( \frac{2}{\mu} \sqrt{\lambda(e^{\mu r_0} - e^{\mu r})(e^{\mu s} - e^{\mu s_0})} \right). \tag{38}$$

Similarly, letting $\mu \to 0$, the Telegrapher's Equation (30), is recovered.

Apparently unaware of the work of Cohn, in 1986, Vaz et al. [49] proved an existence and uniqueness theorem for the iterative process (33). They considered the more general problem

$$U_{rs} - f(r,s)U = 0, \tag{39}$$

where

$$
\begin{aligned}
U(r,s_0) &= \sigma(r), \\
U(r_0,s) &= \tau(s), \\
U(r_0,s_0) &= \sigma(r_0) = \tau(s_0).
\end{aligned}
$$

Reformulate Equation (39) in terms of the linear Volterra integral equation

$$U(r,s) = g(r,s) + \int_{B[r,s]} f(\alpha,\beta)U(\alpha,\beta)d\alpha d\beta \tag{40}$$

for which $g(r,s) = \sigma(r) + \tau(s) - \sigma(r_0)$ and $B[r,s] = [r_0,r] \times [s_0,s]$, as defined by Cohn. Writing the solution of (40) as

$$
\begin{aligned}
U(r,s) &= \lim_{k} \left\{ g(r,s) + \int_{B[r,s]} f(\alpha_1,\beta_1)g(\alpha_1,\beta_1) + \dots \right\} \\
&= \int_{B[r,s]} \dots \int_{B[r_{k-1},s_{k-1}]} f(\alpha_1,\beta_1) \dots f(\alpha_k,\beta_k)g(\alpha_k,\beta_k)d\alpha_k d\beta_k \dots d\alpha_1 d\beta_1
\end{aligned} \tag{41}
$$

the existence and uniqueness of (41) was then proven. Cohn chose a specific functional form for $f(r,s)$. In principle, the approach can be applied to more general functions - hence the usefulness of the theorem. Vaz et al. subsequently used (41) to derive the Riemann function for the Telegrapher's Equation (30).

### 3.3. Olevskiĭ's Addition Formula

Copson's method three from his review paper [24] proposed a way to derive Riemann functions that extends Riemann's own contribution to the field. It relies on the separability of the governing equation. However, it turns out that an addition formula, which encompasses Copson's approach, was already in the Russian literature in 1952 [42]. This was not noted until 1977, in an erratum [50] to Papadakis and Wood's paper [51], which rederived Olevskiĭ's result.

In Olevskiĭ's notation, the Riemann function, $R_{\rho_1 - \rho_2}$, of the equation

$$U_{yy} - U_{xx} + (\rho_1(y) - \rho_2(x))\,U = 0, \tag{42}$$

is

$$R_{\rho_1 - \rho_2}(x,y,x_0,y_0) = R_{\rho_2}(x,y,x_0,y_0) + \int_{y-y_0}^{x-x_0} R_{\rho_2}(x,t,x_0,0)\frac{\partial}{\partial t}R_{\rho_1}(t,y,0,y_0)dt. \tag{43}$$

$R_{\rho_1}$ and $R_{\rho_2}$ are the Riemann functions for

$$U_{yy} - U_{xx} + \rho_1(y)U = 0$$

and

$$U_{yy} - U_{xx} - \rho_2(x)U = 0.$$

Alternatively, integrating by parts yields the equivalent formula

$$R_{\rho_1 - \rho_2}(x, y, x_0, y_0) = R_{\rho_1}(x, y, x_0, y_0) + \int_{x - x_0}^{y - y_0} R_{\rho_1}(t, y, 0, y_0) \frac{\partial}{\partial t} R_{\rho_2}(x, t, x_0, 0) dt. \tag{44}$$

Olevskiĭ obtained this result in part by using the method of successive approximations, which mirrors Cohn's approach in Section 3.2, but there are not many details in the paper. Mostly, Olevskiĭ limits himself to showing that (43) satisfies (42) and the conditions (11)–(13).

To apply this addition formula efficiently, the following identity, which was first employed by Chaundy [30], can be useful. Within the region of uniform convergence of the infinite series that define the indicated functions, we have

$$\begin{aligned}
F_B(a_1, &\ldots, a_p, a_1', \ldots, a_q', b_1, \ldots, b_p, b_1', \ldots, b_q', 1, x_1, \ldots, x_p, x_1', \ldots, x_q') \\
&= F_B(a_1, \ldots, a_p; b_1, \ldots, b_p; 1; x_1, \ldots, x_p) \\
&\quad + \int_0^1 F_B(a_1, \ldots, a_p; b_1, \ldots, b_p; 1; (1-t)x_1, \ldots, (1-t)x_p) \\
&\quad \times \frac{d}{dt} F_B(a_1', \ldots, a_q'; b_1', \ldots, b_q'; t x_1', \ldots, t x_q') dt.
\end{aligned} \tag{45}$$

Olevskiĭ included three specific examples in his paper, illustrating the use of the addition formula. We shall consider one of those examples here. Firstly, Olevskiĭ stated that the Riemann function for

$$U_{yy} - U_{xx} + \frac{m(1-m)}{\sin^2 y} U = 0, \tag{46}$$

was

$$R(x, y, x_0, y_0) = {}_2F_1(m, 1 - m; 1; z_7), \tag{47}$$

where

$$z_7 = \frac{\cos(y - y_0) - \cos(x - x_0)}{2 \sin y \sin y_0}. \tag{48}$$

No derivation for this Riemann function was given. In particular, it is not on the list of self-adjoint Equations (21) from Section 2.1. Nevertheless, interchanging the roles of $x$ and $y$ and letting $m \to n$ yields

$$U_{yy} - U_{xx} - \frac{n(1-n)}{\sin^2 x} U = 0, \tag{49}$$

with the associated Riemann function,

$$R(x, y, x_0, y_0) = {}_2F_1(n, 1 - n; 1; z_8), \tag{50}$$

and

$$z_8 = \frac{\cos(x - x_0) - \cos(y - y_0)}{2 \sin x \sin x_0}. \tag{51}$$

Hence applying (44), the Riemann function of

$$U_{yy} - U_{xx} + \left[ \frac{m(1-m)}{\sin^2 y} - \frac{n(1-n)}{\sin^2 x} \right] U = 0 \tag{52}$$

is

$$R(x, y, x_0, y_0) = {}_2F_1(m, 1-m, 1, z_7) + \int_{x-x_0}^{y-y_0} {}_2F_1(m, 1-m, 1, u_1(t)) \frac{\partial}{\partial t} {}_2F_1(n, 1-n, 1, v_1(t)) dt, \tag{53}$$

where

$$u_1(t) = \frac{\cos(y-y_0) - \cos(t)}{2 \sin y \sin y_0}, \qquad v_1(t) = \frac{\cos(x-x_0) - \cos(t)}{2 \sin x \sin x_0}.$$

Now make the change of variables

$$t' = \frac{\cos(x-x_0) - \cos(t)}{\cos(x-x_0) - \cos(y-y_0)}$$

so that (53) becomes

$$R(x, y, x_0, y_0) = {}_2F_1(m, 1-m, 1, z_7) + \int_0^1 {}_2F_1(m, 1-m, 1, (1-t')z_7) \frac{\partial}{\partial t'} {}_2F_1(n, 1-n, 1, z_8 t') dt'.$$

Using (45), this is recognised to be

$$R(x, y, x_0, y_0) = F_3(m, 1-m, n, 1-n; 1; z_7, z_8), \tag{54}$$

where $z_7$ and $z_8$ are defined by (48) and (51).

Equations (46) and (52) are in fact subcases of (17). This will be explored further in Sections 4.1 and 4.5. Again, this shows the historic difficulty in identifying equivalences in the literature.

## 4. Developments Since 1958

Several new constructive techniques have been proposed since the publication of Copson's paper. The resulting Riemann functions are often equivalent, under a change of variables, although that fact has often been missed in the literature. The aim here is to outline the solution methods and clarify the equivalences. A key finding is that the equivalence can be defined in terms of the governing equation admitting a symmetry algebra isomorphic to $SL(2, R)$ or to a Lie-Bäcklund symmetry algebra.

### 4.1. Lie Point Symmetries

As stated in Section 3.1, the use of symmetry groups to find Riemann functions dates to 1937. However, a modern treatment of the technique did not emerge until 1967, in the Ph.D. thesis of Bluman [44,52]. Subsequently, the approach has become an active area of investigation [45,46,53–55]. Bluman's method gave a completely algorithmic way of applying Lie point symmetries to Riemann's method and thus further developed the technique pioneered by Courant and Hilbert.

Bluman's treatment employed the (now) well-understood infinitesimal representation for deriving symmetry reductions. Employing the definitions and terminology of Olver [56], let

$$v = \sum_{i=1}^p \xi^i(\mathbf{x}, \mathbf{u}) \frac{\partial}{\partial x^i} + \sum_{\alpha=1}^q \phi_\alpha(\mathbf{x}, \mathbf{u}) \frac{\partial}{\partial u^\alpha}, \tag{55}$$

where

$$
\begin{aligned}
\mathbf{x} &= (x^1, x^2, x^3, \ldots, x^p) \equiv (x, y, z, \ldots), \\
\mathbf{u} &= (u^1, u^2, \ldots, u^q) \equiv (u, u_x, u_y, u_z, \ldots),
\end{aligned}
$$

be a vector field defined on an open subset, $M \subset X \times U$, of the space of independent and dependent variables. The $n$-th prolongation of $v$ is the vector field,

$$
Pr^{(n)}v = v + \sum_{\alpha=1}^{q} \sum_{J} \phi_{\alpha}^{J}(\mathbf{x}, \mathbf{u}^{(\mathbf{n})}) \frac{\partial}{\partial u_{J}^{\alpha}}, \tag{56}
$$

defined on the corresponding jet space, $M^{(n)} \subset X \times U^{(n)}$, with the second summation being over all unordered multi-indices $J = (j_1, \ldots, j_k)$ and $1 \leq j_k \leq p, 1 \leq k \leq n$. The coefficient functions $\phi_{\alpha}^{J}$ of $Pr^{(n)}v$ are given by the following formula:

$$
\phi_{\alpha}^{J}(x, u^{(n)}) = D_{j}\left(\phi_{\alpha} - \sum_{i=1}^{p} \xi^{i} u_{i}^{\alpha}\right) + \sum_{i=1}^{p} \xi^{i} u_{J,i}^{\alpha}. \tag{57}
$$

The total differential, $D_j$, is given by

$$
D_{j} = \frac{\partial}{\partial x^{i}} + \sum_{\alpha=1}^{q} \sum_{J} u_{J,i}^{\alpha} \frac{\partial}{\partial u_{J}^{\alpha}}.
$$

We also have that

$$
u_{i}^{\alpha} = \frac{\partial u^{\alpha}}{\partial x^{i}}, \qquad u_{J,i}^{\alpha} = \frac{\partial u_{J}^{\alpha}}{\partial x^{i}} = \frac{\partial^{k+1} u^{\alpha}}{\partial x^{i} \partial x^{j_1} \cdots \partial x^{j_k}}.
$$

Now suppose that

$$
\Delta(x, u^{(n)}) = 0
$$

is an $n$-th order system of differential equations of maximal rank defined over $M \subset X \times U$. If $G$ is a local group of transformations acting on $M$, and

$$
Pr^{(n)}v[\Delta(x, u^{(n)})] = 0, \quad \text{whenever} \quad \Delta(x, u^{(n)}) = 0, \tag{58}
$$

for every infinitesimal generator $v$ of $G$, then $G$ is a symmetry group of the system.

Taking these definitions, Bluman analysed the EPD equation written as

$$
U_{yy} - U_{xx} - \frac{2m}{x} U_{x} = 0, \tag{59}
$$

which is slightly different to (14) (Any equation $U_{yy} - U_{xx} - f(x)U_x = 0$, where $f(x)$ is an arbitrary function, may be put into self-adjoint form by the transformation $U = e^{-\frac{1}{2} \int^{x} f(t)dt} V(x,y)$ to obtain $V_{yy} - V_{xx} + \frac{1}{2}\left(f' + \frac{1}{2}f^2\right)V = 0$. Hence, the change of variable $U = x^{-m}V(x,y)$ will convert (59) to (14).)

Bluman's idea was to follow the 1962 lead of Mackie [57] and find the Green's function of (59). Mackie showed that the relationship between the Riemann and Green's functions can be written as

$$
R(x, y, x_0, y_0) = \begin{cases} -2G(x_0, y_0, x, y) & \text{if } (x, y) \text{ lies inside } P_0 AB \\ 0 & \text{if } (x, y) \text{ lies outside } P_0 AB \end{cases} \tag{60}
$$

where $P_0AB$ was defined in Figure 1.

So given the vector field,

$$v = \xi(x,y)\frac{\partial}{\partial x} + \eta(x,y)\frac{\partial}{\partial y} + \alpha(x,y)U\frac{\partial}{\partial U},$$

the symmetry group of (59) is found by taking

$$Pr^{(2)}v[U_{yy} - U_{xx} - \frac{2m}{x}U_x - \delta(x - x_0)\delta(y - y_0)] = 0$$

to obtain a system of seven determining equations for the infinitesimals $\xi$, $\eta$, and $\alpha$. Solving these equations produces a four-dimensional symmetry group. The presence of the delta functions (and therefore of their derivatives) gives rise to the extra condition $\xi(x_0, y_0) = \eta(x_0, y_0) = 0$. Applying this condition, Bluman obtained the specific group

$$\begin{aligned}
\xi &= -\frac{x}{m}(y - y_0), \\
\eta &= \frac{(x_0^2 - x^2) - (y - y_0)^2}{2m}, \\
\alpha &= (y - y_0).
\end{aligned}$$

Hence

$$\frac{dU}{(y - y_0)U} = -\frac{mdx}{x(y - y_0)} = \frac{2mdy}{(x_0^2 - x^2) - (y - y_0)^2}.$$

The similarity variables are defined by the integrals

$$\begin{aligned}
\frac{dx}{dy} &= \frac{2x(y - y_0)}{(y - y_0)^2 + (x^2 - x_0^2)}, \\
\frac{dU}{U} &= -\frac{mdx}{x},
\end{aligned}$$

which give

$$\begin{aligned}
U(x,y) &= x^{-m}V(z), \\
z &= x - \frac{(y - y_0)^2}{x} + \frac{x_0^2}{x},
\end{aligned} \tag{61}$$

with $V(z)$ an arbitrary function of $z$. Substituting (61) into (59), Bluman found that

$$(z^2 - 4x_0^2)V'' + 2zV' + m(1 - m)V = 0. \tag{62}$$

Letting $\psi = (2x_0 - z)/(4x_0)$, (62) becomes the hypergeometric equation

$$\psi(1 - \psi)\frac{d^2V}{d\psi^2} + (1 - 2\psi)\frac{dV}{d\psi} - m(1 - m)V = 0.$$

The solution displaying the desired properties, (3)–(6), is $V(\psi) = A\,_2F_1(m, 1 - m; 1; \psi)$, where $A$ is a normalization constant to be determined. Putting this all together and using (60), the Riemann function for (59) is then

$$R(x, y, x_0, y_0) = \left(\frac{x}{x_0}\right)^m \,_2F_1\left(m, 1 - m; 1; \frac{(y - y_0)^2 - (x - x_0)^2}{4xx_0}\right). \tag{63}$$

In 1970, Daggit [45] also employed symmetry groups in an effort to find Riemann functions. His idea was to obtain conditions on the coefficients $a$, $b$, and $c$ of (1) which allow a similarity reduction to the Riemann function. Daggit found three distinct cases in which the Riemann function could be found. Kokinasidi [53] also presented similar results to those given by Daggit. Extending Daggit's results, Wood [58] defined any Riemann function

$$R(r, s, r_0, s_0) = M(r, s, r_0, s_0) f(G(r, s, r_0, s_0), r_0, s_0),$$

where $M$ and $G$ are functions, to be a *simple* Riemann function when (i) the equation is almost self-adjoint, (ii) the ODE for $f$ has coefficients depending on $G$, and (iii) the characteristic conditions for the Riemann function become initial conditions for $f$ (Wood's assumed form of the solution, when taken in light of the fact that he is essentially discussing similarity solutions, can be thought of as a first step towards the direct method of symmetry calculation derived by Clarkson and Kruskal [59] in 1989). All the results of Daggit [45], Riemann [25] in Section 2 method 1 and Cohn [41] from Section 3.2 are simple Riemann functions. Wood concluded that, for self-adjoint equations, all the simple Riemann functions had been found by these three authors. Geddes and Mackie [60] arrived at essentially the same conclusion when they found Cohn's method [41] does not produce any new results even if one seeks a Riemann function $R(z; r_0, s_0)$ where $R$ satisfies an ODE.

In other words, the equations

$$U_{rs} + U = 0, \tag{64}$$

$$U_{rs} - \frac{m(1-m)}{(r-s)^2} U = 0, \tag{65}$$

$$U_{rs} - \frac{m(1-m)}{\sinh^2(r-s)} U = 0, \tag{66}$$

$$U_{rs} - \frac{m(1-m)}{\sin^2(r-s)} U = 0, \tag{67}$$

$$U_{rs} + \frac{m(1-m)}{\cosh^2(r-s)} U = 0, \tag{68}$$

are essentially the only possibilities. The first three are the Telegrapher's equation, the EPD equation and Cohn's equation. The fourth equation is (46) from Section 3.3 that was cited, without derivation, by Olevskiĭ. The last one appears to be new.

Wood's statement that these are the only possible cases needs clarifying. In [61], Bluman showed how to construct invertible point transformations, which map a given PDE into another PDE, in the sense that any solution of the given PDE is mapped into a solution of the target PDE. In general, such a mapping need not be a group transformation. Although, if the mapping from a given PDE to a target PDE is one-to-one (invertible) then the transformation must establish a one-to-one correspondence between the infinitesimal generators of the given PDE and the target PDE. More precisely, it is necessary that any given Lie algebra of infinitesimal generators of the given PDE be isomorphic to a Lie algebra of infinitesimal generators of the target PDE. Recall that a Lie algebra $\mathcal{L}_x$ with basis set $\{X_1, X_2, \ldots, X_n\}$ will be isomorphic to the Lie algebra $\mathcal{L}_z$ with basis set $\{Z_1, Z_2, \ldots, Z_n\}$ if

$$[Z_\alpha, Z_\beta] = C_{\alpha\beta}^\gamma Z_\gamma \tag{69}$$

and

$$[X_\alpha, X_\beta] = C_{\alpha\beta}^\gamma X_\gamma, \tag{70}$$

where

$$[X_\alpha, X_\beta] = X_\alpha X_\beta - X_\beta X_\alpha \tag{71}$$

is the usual commutator and the structure constants $\{C^\gamma_{\alpha\beta}\}$ are the same for $\mathcal{L}_x$ and $\mathcal{L}_z$. A few calculations show that the symmetry algebras of (65)–(68) satisfy (69)–(71) and (64) is obtained via a confluence. In effect, (64)–(68) form an equivalence class.

In 1991, Ibragimov [46], unaware of the research of Daggit et al., published a unifying result connecting all previous symmetry group studies for Riemann's method. Again, Ibragimov looked at predicting the form of the Riemann function based on the coefficients $a$, $b$, and $c$ of (1). The idea is that any linear hyperbolic PDE can be classified based on the size of the group that it admits. Ovsiannikov [62] showed that (1) admits a four-dimensional Lie algebra, if and only if, the invariants (Ovsiannikov invariants)

$$p = \frac{k}{h}, \qquad q = \frac{\ln |h|_{rs}}{h}, \tag{72}$$

are constants (if $h = 0$ then swap $k$ and $h$). When $p$ or $q$ (or both of them) is not constant then the symmetry algebra is two dimensional. Here, $h$ and $k$ are Laplace invariants of (1)

$$h = a_r + ab - c, \qquad k = b_s + ab - c.$$

When $p$ and $q$ are constant and $q \neq 0$, one obtains case one as found by Daggit. Whereas Daggit's cases two and three correspond to $q = 0$.

As calculated by Bluman, the symmetry algebra of the EPD equation is four dimensional. Furthermore, the symmetry algebra of the EPD equation is isomorphic to $SL(2, R)$ [46]. Using standard results from Lie theory [63], the symmetries exponentiate to a local Lie representation of the group $SL(2, R)$ by operators $T(G)$, where

$$T(G)U(t,r) = \left[ (\alpha + \gamma t)^2 - \gamma^2 r^2 \right]^{-1/2} U \left[ \frac{(\delta t + \beta)(\alpha + \gamma t) - \gamma \delta r^2}{(\alpha + \gamma t)^2 - \gamma^2 r^2}, \frac{r}{(\alpha + \gamma t)^2 - \gamma^2 r^2} \right]$$

and

$$G = \begin{pmatrix} \alpha & \beta \\ \gamma & \delta \end{pmatrix} \in SL(2, R).$$

Using this fact, one can show that the fundamental solution can be represented in terms of the hypergeometric function. In effect, all the Riemann functions given in this section are encompassed by this result.

In 2015, Andrey et al. [55] studied the equation

$$U_{rs} - \frac{1}{2r} U_s + sU = 0, \tag{73}$$

for which the Riemann function was obtained via a symmetry reduction as

$$R(r, s, r_0, s_0) = \sqrt{\frac{r_0}{r}} J_0 \left( \sqrt{2(r - r_0)(s^2 - s_0^2)} \right). \tag{74}$$

Applying (72) shows that $p = 1$ and $q = 0$. Hence, (73) falls into case two of Daggit and it admits a four dimensional symmetry algebra. Equation (73) is simply a confluent form of the EPD Equation (14), under a transformation of variables.

In 2003, an interesting qualitative property linked to (72) was published in [54]. There, it was shown that if $k = 0$ and $q$ is a non-zero constant in (72), then (1) can be written as

$$U_{rs} + \frac{1}{2}U_r - \frac{2}{q}\frac{1}{r+s}U_t - \frac{1}{q}\frac{1}{r+s}U = 0, \tag{75}$$

which can be factorized to give

$$\left( \frac{\partial}{\partial r} - \frac{2}{q}\frac{1}{r+s} \right)\left( \frac{\partial}{\partial s} + \frac{1}{2} \right)U = 0. \tag{76}$$

The Riemann function is then easily obtained from the factorization.

### 4.2. Laplace Transform for a Klein-Gordon Equation with a Non-Constant Coefficient

The first paper exploring the application of Laplace transforms to Riemann's method was due to Scott [64]. In 1977, Wahlberg [65] also used the technique to find the Riemann function for

$$U_{yy} - U_{xx} + (1 + \delta x)U = 0. \tag{77}$$

To solve (77), first make the change of variables,

$$r = \frac{1}{2}\lambda^{1/3}\left[ (x_0 + t_0) - (x + t) \right], \qquad s = \frac{1}{2}\lambda^{1/3}\left[ (x - t) - (x_0 - t_0) \right]. \tag{78}$$

Equation (77) then reduces to

$$\frac{\partial^2 W}{\partial r \partial s} + (\sigma + s - r)W = 0, \tag{79}$$

$$W = 1 \qquad \text{when} \qquad rs = 0, \tag{80}$$

where $\sigma = \lambda^{-2/3}(1 + \lambda x_0)$.

Note that from (72), $p = 1$ but $q \neq const$. Hence, (79) is not part of the equivalence class of equations admitting a symmetry group isomorphic to $SL(2, R)$ that was seen in Section 4.1. By extension, there is no invertible transformation from (79) to the EPD Equation (14).

Equation (79) is then solved by Laplace transform. Write the transform of $W(r, s)$ as

$$\tilde{W}(r, p) \equiv \int_0^\infty e^{-ps}W(r, s)ds.$$

The transformed version of (79) is

$$p\frac{\partial \tilde{W}}{\partial r} - \frac{\partial \tilde{W}}{\partial p} + (\sigma - r)\tilde{W} = 0 \tag{81}$$

after applying the boundary condition (80). The solution to (81) is easily obtained as

$$\tilde{W}(r, p) = F\left( r + \frac{1}{2}p^2 \right)\exp\left[ (\sigma - r)p - \frac{1}{3}p^3 \right]. \tag{82}$$

The function $F$ is determined from the condition that $W(0, s) = 1$, or $\tilde{W}(0, p) = 1/p$, which yields

$$\tilde{W}(r, p) = \frac{1}{\sqrt{2r + P^2}}\exp\left[ (\sigma - r)p - \frac{1}{3}p^3 - \sigma\sqrt{2r + P^2} + \frac{1}{3}\left( 2r + p^2 \right)^{3/2} \right]. \tag{83}$$

Let $p = \xi + i\eta$, the solution to (79) and (80) is then formally given by

$$W(r, s) = \frac{1}{2\pi i} \int_{\xi_0 - i\infty}^{\xi_0 + i\infty} \tilde{W}(r, p) dp, \tag{84}$$

where $\tilde{W}(r, p)$ was defined in (83).

Expressing (84) as a contour integral, Wahlberg showed that the Riemann function for (77) can then be represented by

$$R(\xi, \eta, \xi_0, \eta_0) = \frac{1}{2\pi i} \int_\Gamma \exp\left\{ \left[1 + \frac{\delta}{2}(\xi + \xi_0)\right] z + \frac{(\xi - \xi_0)^2 - (\eta - \eta_0)^2}{4z} - \frac{\delta^2}{12} z^3 \right\} \frac{dz}{z}, \tag{85}$$

where $\Gamma$ is defined by $|z| = \rho$, encircling the origin in the positive direction (since the integrand is analytic everywhere except at $z = 0$, $\Gamma$ can be taken to be any closed contour encircling the origin in the positive direction).

Alternatively, expanding $\exp(-\delta^2 z^3 / 12)$ of the integrand as a power series then applying the formula [66]

$$J_n(\alpha) = \frac{1}{2\pi i} \int_\Gamma z^{-n-1} \exp\left[\frac{1}{2}\alpha \left(z - \frac{1}{z}\right)\right] dz$$

for the Bessel function of order $n$, yields

$$R(x, y, x_0, y_0) = \sum_{n=0}^\infty \frac{\Omega^n}{n!} J_{3n}(\Delta), \tag{86}$$

with

$$\Omega = \frac{\delta^2}{96} \left( \frac{(y - y_0)^2 - (x - x_0)^2}{1 + \frac{\delta}{2}(x + x_0)} \right)^{3/2}, \tag{87}$$

$$\Delta = \left\{ \left[1 + \frac{\delta}{2}(x + x_0)\right] \left[(y - y_0)^2 - (x - x_0)^2\right] \right\}^{1/2}. \tag{88}$$

### 4.3. The Multiplication Formula

In 1981, Xin Hua Du [67] derived a multiplication theorem for Riemann functions. Returning to the method of successive approximations, first used by Cohn [41] and Olevskiĭ [42] as outlined in Section 3, Du wrote the Riemann function for the equation

$$U_{yy} - U_{xx} + A(x, y)U = 0 \tag{89}$$

as

$$\begin{aligned} R_A(x, y, x_0, y_0) &= \sum_{k=1}^\infty G_k(\xi, \eta; \xi_0, \eta_0) \\ &= \sum_{k=1}^\infty \left(-\frac{1}{4}\right)^{k-1} \int_{Q_0}^Q \int_{Q_0}^{Q_{k-1}} \cdots \int_{Q_0}^{Q_1} \prod_{i=1}^{k-1} A(Q_i) dQ_1 dQ_2 \ldots dQ_{k-1}, \end{aligned} \tag{90}$$

where $G_1 = 1$ and

$$\int_{\xi_0}^\xi \int_{\eta_0}^\eta A\left(\frac{p_1 + q_1}{2}, \frac{p_1 - q_1}{2}\right) dp_1 dq_1 = \int_{Q_0}^Q A(Q_1) dQ_1.$$

For simplicity write

$$R_A = \sum T_K(A).$$

From this starting point, Du supposed that $A(x,y) = \rho(x)$ in (89). In this way, the corresponding Riemann function (90) can be denoted by $R_\rho(x, y, x_0, y_0)$. Then the Riemann function, $R_{\ln \rho}$, for the equation

$$U_{yy} - U_{xx} + (\ln \rho(x)) U = 0 \tag{91}$$

is

$$R_{\ln \rho}(x, y, x_0, y_0) = [R_\rho]^*_{\ln \rho} = \sum T_k \left( \ln \frac{-(\partial_{yy} - \partial_{xx}) R_\rho}{R_\rho} \right). \tag{92}$$

Conversely, if $R_{\ln \rho}$ is given, then similarly we have

$$R_\rho(x, y, x_0, y_0) = [R_{\ln \rho}]^\rho_* = \sum T_k \left( e^{-\frac{(\partial_{yy} - \partial_{xx}) R_{\ln \rho}}{R_{\ln \rho}}} \right). \tag{93}$$

Again for convenience, write $[R_\rho]^*_{\ln \rho}$ and $[R_{\ln \rho}]^\rho_*$ as $[R_\rho]^*$ and $[R_{\ln \rho}]_*$. We are now in a position to state Du's main result. If $\rho_1 = \rho_1(x)$ and $\rho_2 = \rho_2(y)$, then the Riemann function, $R_{\rho_1 \rho_2}$, for the equation

$$U_{yy} - U_{xx} + [\rho_1(x)\rho_2(y)]U = 0 \tag{94}$$

can be given by

$$R_{\rho_1 \rho_2}(x, y, x_0, y_0) = \left[ (R_{\rho_1 + \rho_2}) \circ (R_{\rho_1} = [R_{\rho_1}]^*, R_{\rho_2} = [R_{\rho_2}]^*) \right]_*, \tag{95}$$

where $(\quad) \circ (\quad)$ represents a convolution. Furthermore $R_{\rho_1}$ and $R_{\rho_2}$ must possess the properties

$$R_{\rho_1}(x, y, x_0, y_0) = R_{\rho_1}(x, y - y_0, x_0, 0) = R_{\rho_1}(x, y_0 - y, x_0, 0), \tag{96}$$
$$R_{\rho_2}(x, y, x_0, y_0) = R_{\rho_2}(x - x_0, y, 0, y_0) = R_{\rho_2}(x_0 - x, y, 0, y_0). \tag{97}$$

The above notation for (91)–(95) is exactly that used by Du. However it is somewhat cryptic. So a little explanation may be useful. What Du denotes by $[\quad]_*$ in (95) is essentially the Riemann function for

$$U_{yy} - U_{xx} + \ln[\rho_1(x)\rho_2(y)] U = 0. \tag{98}$$

This is built using Olevskiĭ's addition Formula (43) and the equations

$$U_{yy} - U_{xx} + \ln \rho_1(x) U = 0,$$
$$U_{yy} - U_{xx} + \ln \rho_2(y) U = 0,$$

for which the required Riemann functions can be obtained from (92). Then applying (93) to (98) yields the required Riemann function for (94).

*4.4. Finite Groups and a Link to Appell's $F_4$*

First, return to the equation

$$L[U] = U_{rs} + \left[ \frac{m(m-1)}{(r-s)^2} - \frac{n(n-1)}{(r+s)^2} \right] U = 0. \tag{99}$$

Recall that it was seen as case 4 of (21), where it was obtained as a reduction of Chaundy's Equation (17). The Riemann function in characteristic variables was

$$R(r,s,r_0,s_0) = F_3(n, 1-n, m, 1-m; 1; z_1, z_2),$$

where $z_1$ and $z_2$ are given by (19). Applying (72), it is straight forward to show that the Riemann function for (99) cannot be obtained by means of point symmetries as the group is trivial. However, Iwasaki [47] found in 1988 that, although a symmetry reduction will not yield the Riemann function, a finite group isomorphic to $\mathbb{Z}_2 \times \mathbb{Z}_2 \times \mathbb{Z}_2$ acting on its Riemann function permitted him to reduce (99) to Appell's system $F_4$.

As Iwasaki showed, the Riemann function $R(r,s,r_0,s_0)$ is invariant under the transformations of the independent variables of the form

1.　$(r,s,r_0,s_0) \mapsto (\lambda r, \lambda s, \lambda r_0, \lambda s_0)$,　$\lambda$　constant,
2.　$(r,s,r_0,s_0) \mapsto (1/r, 1/s, 1/r_0, 1/s_0)$,
3.　$(r,s,r_0,s_0) \mapsto (s,r,s_0,r_0)$,
4.　$(r,s,r_0,s_0) \mapsto (r_0,s_0,r,s)$.

Taking advantage of transformation 1, define the new variables

$$X = \frac{r}{s_0}, \quad Y = \frac{s}{s_0}, \quad Z = \frac{r_0}{s_0}. \tag{100}$$

The Riemann function must now satisfy

$$L_{(X,Y)}[R] = 0, \tag{101}$$

$$(X-Z)(Y-1) = 0 \quad \text{implies} \quad R = 1. \tag{102}$$

The group $G$ generated by the transformations of $(X,Y,Z)$ have the generators

$$X^* : \quad (X,Y,Z) \mapsto (X, X/Z, X/Y),$$
$$Y^* : \quad (X,Y,Z) \mapsto (Y/Z, Y, Y/X),$$
$$Z^* : \quad (X,Y,Z) \mapsto (Z/Y, Z/X, Z).$$

As is easily seen, $X^*$, $Y^*$, and $Z^*$ are involutions and commutative with each other. Hence the group $G = < X^*, Y^*, Z^* >$ is $G \approx \mathbb{Z}_2 \times \mathbb{Z}_2 \times \mathbb{Z}_2$ and $|G| = 8$.

Iwasaki then sought an extension to the field $\mathbf{K}/\mathbf{k}$ where $\mathbf{K} = C(X,Y,Z)$ and $\mathbf{k} = \{f \in \mathbf{K}; f \text{ is } G\text{-invariant}\}$. With this in mind, he defined the new variables

$$p_i = X_i + 1/X_i + X_k/X_j + X_j/X_k, \tag{103}$$
$$q_i = X_i/(X_jX_k) + (X_jX_k)/X_i + 2, \tag{104}$$
$$s_i = X_i + 1/X_i - X_k/X_j - X_j/X_k, \tag{105}$$

where $(i,j,k)$ runs over all permutations of $(1,2,3)$ and $X_1 = X$, $X_2 = Y$ and $X_3 = Z$. Thus $p_i, q_i \in \mathbf{k}$. After further examination of the field, Iwasaki was led to try

$$r' = p_1/q_3, \quad s' = p_2/q_3, \quad t' = q_3. \tag{106}$$

Substitution of (106) into (101) and (102) gives

$$\{s_3 M_1 + r's't' M_1 + 2M_2 + 2M_3\} R = 0, \tag{107}$$

where

$$
\begin{aligned}
M_1 &= L_{(r',s')} = \partial_{r'}\partial_{s'} + \frac{n(1-n)}{(r'-s')^2} - \frac{m(1-m)}{(r'+s')^2}, \\
M_2 &= (1-r'^2)\partial_{r'}^2 + (1-s'^2)\partial_{s'}^2 - 2r's'\partial_{r's'}^2 - 2r'\partial_{r'} - 2s'\partial_{s'} - \frac{2n(1-n)}{(r'-s')^2} - \frac{2m(1-m)}{(r'+s')^2}, \\
M_3 &= t'\{(t'-4)\partial_{t'}(t'\partial_{t'} - r'\partial_{r'} - s'\partial_{s'} - 1) + 2\partial_{t'}\},
\end{aligned}
$$

and $s_3$ is defined by (105). Since $M_1 R$ and $r's't' M_1 R + 2M_2 R + 2M_3 R$ are $G$-invariant and $s_3$ is not an element of **k**, (107) splits into two parts:

$$M_1 R = 0, \qquad M_2 R + M_3 R = 0. \tag{108}$$

Iwasaki then proposed that if $f(r', s', t')$ is a solution of (108) (suppose that $f$ makes sense at $t' = 0$), then $f(r', s', 0)$ is also a solution of (108). Hence it is reasonable to expect that $R(r, s, r_0, s_0)$ is a function depending only on $(r', s')$ and to consider a system of PDEs

$$M_1 U = M_2 U = 0, \tag{109}$$

to which (108) reduces if this expectation is correct. Now from (102), $s = 1$ implies $u = 1$. If (109) has a solution satisfying this condition then the assumption is correct and $R$ will be given by such a solution. Unfortunately such an expectation is probably specific to this example and may prevent the possibility of building a general method based on Iwasaki's work. In fact, it is reminiscent of the type of inspired guess for which many classical Riemann function papers have been criticised.

In any event, the substitutions

$$
\begin{aligned}
p &= \left[\frac{r'-s'}{2}\right]^2 = \left[\frac{(r-s)(r_0-s_0)}{2(rs+r_0 s_0)}\right]^2, \\
q &= \left[\frac{r'+s'}{2}\right]^2 = \left[\frac{(r+s)(r_0+s_0)}{2(rs+r_0 s_0)}\right]^2, \\
u &= p^{n/2}q^{m/2}v,
\end{aligned}
$$

transform (109) into a system of partial differential equations associated with Appell's hypergeometric function $F_4(\alpha, \beta, \gamma, \gamma'; p, q)$, where $\alpha$ etc., depend on $m$ and $n$. Iwasaki then shows that the general solution can be expressed as a linear combination of four contiguous $F_4$ functions. After further calculations, he concludes that the Riemann function for (99) is

$$R(r, s, r_0, s_0) = \sum_{j=0}^{3} C_j(a, b) u_j(p, q; n, m),$$

where

$$
\begin{aligned}
u_0 &= u(p, q; n, m), & C_0 &= C(n, m), \\
u_1 &= u(p, q; 1-n, m), & C_1 &= C(1-n, m), \\
u_2 &= u(p, q; n, 1-m), & C_2 &= C(n, 1-m), \\
u_3 &= u(p, q; 1-n, 1-m), & C_3 &= C(1-n, 1-m),
\end{aligned}
$$

and

$$u(p,q;n,m) = p^{n/2}q^{m/2}F_4\left(\frac{m+n}{2}, \frac{m+n+1}{2}, n+\frac{1}{2}, m+\frac{1}{2}; p, q\right),$$

$$C(n,m) = \frac{2^{-m-n}\Gamma\left(\frac{1}{2}-n\right)\Gamma\left(\frac{1}{2}-m\right)}{\pi\Gamma\left(1-m-n\right)}.$$

### 4.5. Lie-Bäcklund Symmetries

In 2004, Zeitsch [68,69] was able to establish a new equivalence class of Riemann functions by extending the point symmetry ideas of Section 4.1 to Lie-Bäcklund symmetries. The key result was given in [68]. There it was shown that Chaundy's Equation (17) admitted a two-dimensional generalized symmetry algebra. A straightforward calculation shows that (17) admits no non-trivial point symmetries. Following [70], define the second-order operator

$$S = f_1\partial_{rr} + f_2\partial_{ss} + f_3\partial_r + f_4\partial_s + f_5. \tag{110}$$

We say that (110) is a Lie-Bäcklund symmetry operator for (17) provided

$$[S, L] = QL, \tag{111}$$

where

$$L = \partial_{rs} + \left[\frac{m_1(1-m_1)}{(r+s)^2} - \frac{m_2(1-m_2)}{(r-s)^2} + \frac{m_3(1-m_3)}{(1-rs)^2} - \frac{m_4(1-m_4)}{(1+rs)^2}\right] \tag{112}$$

and

$$Q = h_1\partial_r + h_2\partial_s + h_3 \tag{113}$$

is a first-order differential operator (here Q may vary with S) and $f_1, \ldots, h_3$ are arbitrary functions of $r$ and $s$.

Solving (111) and equating the coefficients of the derivatives to zero yields a two-dimensional vector space of operators. The symmetries, $S_1$ and $S_2$, form a basis and have differential part

$$S_1 = (r^4+1)\partial_{rr} + (s^4+1)\partial_{ss} + 2r^3\partial_r + 2s^3\partial_s, \tag{114}$$

$$S_2 = r^2\partial_{rr} + s^2\partial_{ss} + r\partial_r + s\partial_s. \tag{115}$$

Following [70], for each inequivalent orbit, a separable coordinate system exists. Taking a linear combination of (114) and (115), there are then four orbits, namely:

- $S_1 + 2qS_2, \quad q > 1.$
- $S_1 + 2S_2.$
- $S_1 - 2S_2.$
- $S_1 + 2qS_2, \quad q < -1.$

Analysing each case, Zeitsch then found:
*System 1*: The separable coordinates are

$$r = b\frac{\text{sn}\,[a(\xi+\eta)]}{\text{cn}\,[a(\xi+\eta)]}, \qquad s = b\frac{\text{sn}\,[a(\xi-\eta)]}{\text{cn}\,[a(\xi-\eta)]} \tag{116}$$

where $2q = (1 + b^4)/b^2$, $k^2 = 1 - b^4$, $0 < b < 1$ and $k$ is the modulus of the Jacobian elliptic functions. See [71] for more detail on the elliptic functions sn and cn as well as the other elliptic functions such as dn. When (116) is substituted into (17), the following separable equation is obtained

$$
U_{\xi\xi} - U_{\eta\eta} + \left\{ a^2 \left[ m_1(1 - m_1) \left( \frac{\mathrm{dn}^2 a\xi}{\mathrm{sn}^2 a\xi \mathrm{cn}^2 a\xi} - k^4 \frac{\mathrm{sn}^2 a\eta \mathrm{cn}^2 a\eta}{\mathrm{dn}^2 a\eta} \right) \right. \right.
$$
$$
- m_2(1 - m_2) \left( \frac{\mathrm{dn}^2 a\eta}{\mathrm{sn}^2 a\eta \mathrm{cn}^2 a\eta} - k^4 \frac{\mathrm{sn}^2 a\xi \mathrm{cn}^2 a\xi}{\mathrm{dn}^2 a\xi} \right) \right]
$$
$$
+ 4a^2 b^2 \left[ m_3(1 - m_3) \left( \frac{\mathrm{dn}^2 a\xi}{(\mathrm{cn}^2 a\xi - b^2 \mathrm{sn}^2 a\xi)^2} + \frac{\mathrm{dn}^2 (a\eta)}{(\mathrm{cn}^2 a\eta + b^2 \mathrm{sn}^2 a\eta)^2} - 1 \right) \right.
$$
$$
\left. \left. - m_4(1 - m_4) \left( \frac{\mathrm{dn}^2 a\xi}{(\mathrm{cn}^2 a\xi + b^2 \mathrm{sn}^2 a\xi)^2} + \frac{\mathrm{dn}^2 a\eta}{(\mathrm{cn}^2 a\eta - b^2 \mathrm{sn}^2 a\eta)^2} - 1 \right) \right] \right\} U = 0. \tag{117}
$$

The Riemann function for (117) is found by substituting (116) into (18)–(20). Hence

$$
R(\xi, \eta, \xi_0, \eta_0) = F_B\left(m_1, m_2, m_3, m_4, 1 - m_1, 1 - m_2, 1 - m_3, 1 - m_4, 1, z_{11}, z_{12}, z_{13}, z_{14}\right), \tag{118}
$$

where

$$
z_{11} = \frac{\left[ \begin{array}{c} (\mathrm{sn}\, a\eta \mathrm{cn}\, a\eta_0 \mathrm{dn}\, a\xi_0 - \mathrm{sn}\, a\eta_0 \mathrm{cn}\, a\eta \mathrm{dn}\, a\xi)^2 \\ - (\mathrm{sn}\, a\xi \mathrm{cn}\, a\xi_0 \mathrm{dn}\, a\eta_0 - \mathrm{sn}\, a\xi_0 \mathrm{cn}\, a\xi \mathrm{dn}\, a\eta)^2 \end{array} \right]}{(4\mathrm{sn}\, a\xi\, \mathrm{sn}\, a\xi_0\, \mathrm{cn}\, a\xi\, \mathrm{cn}\, a\xi_0\, \mathrm{dn}\, a\eta\, \mathrm{dn}\, a\eta_0)}, \tag{119}
$$

$$
z_{12} = \frac{\left[ \begin{array}{c} (\mathrm{sn}\, a\xi \mathrm{cn}\, a\xi_0 \mathrm{dn}\, a\eta_0 - \mathrm{sn}\, a\xi_0 \mathrm{cn}\, a\xi \mathrm{dn}\, a\eta)^2 \\ - (\mathrm{sn}\, a\eta \mathrm{cn}\, a\eta_0 \mathrm{dn}\, a\xi_0 - \mathrm{sn}\, a\eta_0 \mathrm{cn}\, a\eta \mathrm{dn}\, a\xi)^2 \end{array} \right]}{(4\mathrm{sn}\, a\eta\, \mathrm{sn}\, a\eta_0\, \mathrm{cn}\, a\eta\, \mathrm{cn}\, a\eta_0\, \mathrm{dn}\, a\xi\, \mathrm{dn}\, a\xi_0)}, \tag{120}
$$

$$
z_{13} = b^2 \frac{\left[ \begin{array}{c} (\mathrm{sn}\, a\eta \mathrm{cn}\, a\eta_0 \mathrm{dn}\, a\xi_0 - \mathrm{sn}\, a\eta_0 \mathrm{cn}\, a\eta \mathrm{dn}\, a\xi)^2 \\ - (\mathrm{sn}\, a\xi \mathrm{cn}\, a\xi_0 \mathrm{dn}\, a\eta_0 - \mathrm{sn}\, a\xi_0 \mathrm{cn}\, a\xi \mathrm{dn}\, a\eta)^2 \end{array} \right]}{\left[ \begin{array}{c} (\mathrm{cn}^2 a\xi - b^2 \mathrm{sn}^2 a\xi)(\mathrm{cn}^2 a\eta + b^2 \mathrm{sn}^2 a\eta) \\ \times (\mathrm{cn}^2 a\xi_0 - b^2 \mathrm{sn}^2 a\xi_0)(\mathrm{cn}^2 a\eta_0 + b^2 \mathrm{sn}^2 a\eta_0) \end{array} \right]}, \tag{121}
$$

$$
z_{14} = b^2 \frac{\left[ \begin{array}{c} (\mathrm{sn}\, a\xi \mathrm{cn}\, a\xi_0 \mathrm{dn}\, a\eta_0 - \mathrm{sn}\, a\xi_0 \mathrm{cn}\, a\xi \mathrm{dn}\, a\eta)^2 \\ - (\mathrm{sn}\, a\eta \mathrm{cn}\, a\eta_0 \mathrm{dn}\, a\xi_0 - \mathrm{sn}\, a\eta_0 \mathrm{cn}\, a\eta \mathrm{dn}\, a\xi)^2 \end{array} \right]}{\left[ \begin{array}{c} (\mathrm{cn}^2 a\xi + b^2 \mathrm{sn}^2 a\xi)(\mathrm{cn}^2 a\eta - b^2 \mathrm{sn}^2 a\eta) \\ \times (\mathrm{cn}^2 a\xi_0 + b^2 \mathrm{sn}^2 a\xi_0)(\mathrm{cn}^2 a\eta_0 - b^2 \mathrm{sn}^2 a\eta_0) \end{array} \right]}. \tag{122}
$$

*System 2:* For the symmetry $S_1 + 2S_2$ we find the separable coordinate system

$$
r = \tan\left[ a \frac{(\xi + \eta)}{2} \right], \qquad s = \tan\left[ a \frac{(\xi - \eta)}{2} \right]. \tag{123}
$$

Substituting (123) into (17) yields

$$
U_{\xi\xi} - U_{\eta\eta} + a^2 \left[ \frac{m_1(1 - m_1)}{\sin^2 a\xi} - \frac{m_2(1 - m_2)}{\sin^2 a\eta} + \frac{m_3(1 - m_3)}{\cos^2 a\xi} - \frac{m_4(1 - m_4)}{\cos^2 a\eta} \right] U = 0. \tag{124}
$$

The Riemann function for (124) is then

$$
R(\xi, \eta, \xi_0, \eta_0) = F_B\left(m_1, m_2, m_3, m_4, 1 - m_1, 1 - m_2, 1 - m_3, 1 - m_4, 1, z_{15}, z_{16}, z_{17}, z_{18}\right), \tag{125}
$$

where

$$z_{15} = \frac{\cos a(\xi - \xi_0) - \cos a(\eta - \eta_0)}{2\sin a\xi \sin a\xi_0}, \quad z_{16} = \frac{\cos a(\eta - \eta_0) - \cos a(\xi - \xi_0)}{2\sin a\eta \sin a\eta_0}, \tag{126}$$

$$z_{17} = \frac{\cos a(\xi - \xi_0) - \cos a(\eta - \eta_0)}{2\cos a\xi \cos a\xi_0}, \quad z_{18} = \frac{\cos a(\eta - \eta_0) - \cos a(\xi - \xi_0)}{2\cos a\eta \cos a\eta_0}. \tag{127}$$

*System 3:* For the symmetry $S_1 - 2S_2$, we find the separable coordinate system

$$r = \tanh\left[a\frac{(\xi + \eta)}{2}\right], \qquad s = \tanh\left[a\frac{(\xi - \eta)}{2}\right]. \tag{128}$$

Substituting (128) into (17) yields

$$U_{\xi\xi} - U_{\eta\eta} + a^2 \left[\frac{m_1(1 - m_1)}{\sinh^2 a\xi} - \frac{m_2(1 - m_2)}{\sinh^2 a\eta} + \frac{m_3(1 - m_3)}{\cosh^2 a\eta} - \frac{m_4(1 - m_4)}{\cosh^2 a\xi}\right] U = 0. \tag{129}$$

The Riemann function for (129) is now

$$R(\xi, \eta, \xi_0, \eta_0) = F_B\left(m_1, m_2, m_3, m_4, 1 - m_1, 1 - m_2, 1 - m_3, 1 - m_4, 1, z_{19}, z_{20}, z_{21}, z_{22}\right) \tag{130}$$

and

$$z_{19} = \frac{\cosh a(\eta - \eta_0) - \cosh a(\xi - \xi_0)}{2\sinh a\xi \sinh a\xi_0}, \quad z_{20} = \frac{\cosh a(\xi - \xi_0) - \cosh a(\eta - \eta_0)}{2\sinh a\eta \sinh a\eta_0}, \tag{131}$$

$$z_{21} = \frac{\cosh a(\eta - \eta_0) - \cosh a(\xi - \xi_0)}{2\cosh a\eta \cosh a\eta_0}, \quad z_{22} = \frac{\cosh a(\xi - \xi_0) - \cosh a(\eta - \eta_0)}{2\cosh a\xi \cosh a\xi_0}. \tag{132}$$

*System 4:* For the symmetry $S = S_1 + 2qS_2$ where $q < -1$, the separable coordinate system is

$$r = b\,\text{sn}\,a(\xi + \eta), \qquad s = b\,\text{sn}\,a(\xi - \eta), \tag{133}$$

where $k = b^2$, $2q = -(1 + b^4)/b^2$ and $0 < b < 1$. As in system 1, $a$ is arbitrary and $k$ is the modulus of the elliptic functions. Substituting (133) into (17) yields

$$
\begin{aligned}
U_{\xi\xi} \quad - \quad & U_{\eta\eta} + \left\{a^2\left[m_1(1 - m_1)\left(\frac{\text{cn}^2 a\xi\,\text{dn}^2 a\xi}{\text{sn}^2 a\xi} - (1 - k^2)^2\frac{\text{sn}^2 a\eta}{\text{cn}^2 a\eta\,\text{dn}^2 a\eta}\right)\right.\right. \\
- \quad & m_2(1 - m_2)\left(\frac{\text{cn}^2 a\eta\,\text{dn}^2 a\eta}{\text{sn}^2 a\eta} - (1 - k^2)^2\frac{\text{sn}^2 a\xi}{\text{cn}^2 a\xi\,\text{dn}^2 a\xi}\right)\right] \\
+ \quad & 4a^2 b^2\left[m_3(1 - m_3)\left(\frac{\text{cn}^2 a\xi\,\text{dn}^2 a\xi}{(1 - b^2\text{sn}^2 a\xi)^2} + \frac{\text{cn}^2 a\eta\,\text{dn}^2 a\eta}{(1 + b^2\text{sn}^2 a\eta)^2} - 1\right)\right. \\
- \quad & \left.\left. m_4(1 - m_4)\left(\frac{\text{cn}^2 a\xi\,\text{dn}^2 a\xi}{(1 + b^2\text{sn}^2 a\xi)^2} + \frac{\text{cn}^2 a\eta\,\text{dn}^2 a\eta}{(1 - b^2\text{sn}^2 a\eta)^2} - 1\right)\right]\right\}U = 0.
\end{aligned} \tag{134}
$$

The Riemann function for (134) is then

$$R(\xi, \eta, \xi_0, \eta_0) = F_B\left(m_1, m_2, m_3, m_4, 1 - m_1, 1 - m_2, 1 - m_3, 1 - m_4, 1, z_{27}, z_{28}, z_{29}, z_{30}\right), \tag{135}$$

where

$$
z_{27} = \frac{\left[ \begin{array}{c} (\operatorname{sn} a\eta \operatorname{cn} a\xi \operatorname{dn} a\xi_0 - \operatorname{sn} a\eta_0 \operatorname{cn} a\xi_0 \operatorname{dn} a\xi)^2 \\ - (\operatorname{sn} a\xi \operatorname{cn} a\eta \operatorname{dn} a\eta_0 - \operatorname{sn} a\xi_0 \operatorname{cn} a\eta_0 \operatorname{dn} a\eta)^2 \end{array} \right]}{(4\operatorname{sn} a\xi \operatorname{sn} a\xi_0 \operatorname{cn} a\eta \operatorname{cn} a\eta_0 \operatorname{dn} a\eta \operatorname{dn} a\eta_0)}, \tag{136}
$$

$$
z_{28} = \frac{\left[ \begin{array}{c} (\operatorname{sn} a\xi \operatorname{cn} a\eta \operatorname{dn} a\eta_0 - \operatorname{sn} a\xi_0 \operatorname{cn} a\eta_0 \operatorname{dn} a\eta)^2 \\ - (\operatorname{sn} a\eta \operatorname{cn} a\xi \operatorname{dn} a\xi_0 - \operatorname{sn} a\eta_0 \operatorname{cn} a\xi_0 \operatorname{dn} a\xi)^2 \end{array} \right]}{(4\operatorname{sn} a\eta \operatorname{sn} a\eta_0 \operatorname{cn} a\xi \operatorname{cn} a\xi_0 \operatorname{dn} a\xi \operatorname{dn} a\xi_0)}, \tag{137}
$$

$$
z_{29} = b^2 \frac{\left[ \begin{array}{c} (\operatorname{sn} a\eta \operatorname{cn} a\xi \operatorname{dn} a\xi_0 - \operatorname{sn} a\eta_0 \operatorname{cn} a\xi_0 \operatorname{dn} a\xi)^2 \\ - (\operatorname{sn} a\xi \operatorname{cn} a\eta \operatorname{dn} a\eta_0 - \operatorname{sn} a\xi_0 \operatorname{cn} a\eta_0 \operatorname{dn} a\eta)^2 \end{array} \right]}{\left[ \begin{array}{c} (1 - b^2 \operatorname{sn}^2 a\xi)(1 + b^2 \operatorname{sn}^2 a\eta) \\ \times (1 - b^2 \operatorname{sn}^2 a\xi_0)(1 + b^2 \operatorname{sn}^2 a\eta_0) \end{array} \right]}, \tag{138}
$$

$$
z_{30} = b^2 \frac{\left[ \begin{array}{c} (\operatorname{sn} a\xi \operatorname{cn} a\eta \operatorname{dn} a\eta_0 - \operatorname{sn} a\xi_0 \operatorname{cn} a\eta_0 \operatorname{dn} a\eta)^2 \\ - (\operatorname{sn} a\eta \operatorname{cn} a\xi \operatorname{dn} a\xi_0 - \operatorname{sn} a\eta_0 \operatorname{cn} a\xi_0 \operatorname{dn} a\xi)^2 \end{array} \right]}{\left[ \begin{array}{c} (1 + b^2 \operatorname{sn}^2 a\xi)(1 - b^2 \operatorname{sn}^2 a\eta) \\ \times (1 + b^2 \operatorname{sn}^2 a\xi_0)(1 - b^2 \operatorname{sn}^2 a\eta_0) \end{array} \right]}. \tag{139}
$$

Returning to (124), setting $m_3 = m_4 = 0$, we recover (52). Hence (52) is in fact a sub-case of Chaundy's equation, obtainable via a Lie-Bäcklund symmetry. A quick calculation shows that (124) only admits one first order symmetry. As such, the group is not isomorphic to $SL(2, R)$. In this way, a new equivalence class of Riemann functions has been established. Likewise, take (129) and let $m_2 = m_3 = 0$. The Riemann function for this case was first given in [51] by using the techniques from Section 3.3. It is therefore also part of the equivalence class admitting the two-dimensional generalized symmetry algebra (114) and (115).

## 5. New Riemann Functions

Based on the link between (124) and (52), seen at the end of Section 4.5, the possibility now exists to construct new Riemann functions, which do not fit into either the point symmetry equivalence class, nor that defined by the generalized symmetries of (17). Taking separable equations and applying them to (43) will result in equations for which there is no invertible transformation to a known Riemann function as the group of the resulting equation will be trivial. Among the separable equations from the previous sections, four possibilities for application to the addition formula arise straight away. If we let $a = \lambda_1$ and $m_1 = m_3 = 0$ in (124), $a = \lambda_2$ and $m_2 = m_3 = 0$ in (129), as well as $(x, y) \to (\eta, \xi)$ in (14) and (77), we obtain

$$
U_{\xi\xi} \quad - \quad U_{\eta\eta} - \lambda_1^2 \left[ \frac{m_2(1 - m_2)}{\sin^2 \lambda_1 \eta} + \frac{m_4(1 - m_4)}{\cos^2 \lambda_1 \eta} \right] U = 0, \tag{140}
$$

$$
U_{\xi\xi} \quad - \quad U_{\eta\eta} + \lambda_2^2 \left[ \frac{m_1(1 - m_1)}{\sinh^2 \lambda_2 \xi} - \frac{m_3(1 - m_3)}{\cosh^2 \lambda_2 \xi} \right] U = 0, \tag{141}
$$

$$
U_{\xi\xi} \quad - \quad U_{\eta\eta} + \frac{n(1 - n)}{\xi^2} U = 0, \tag{142}
$$

$$
U_{\xi\xi} \quad - \quad U_{\eta\eta} + (1 + \delta\xi) U = 0. \tag{143}
$$

The Riemann functions for the first three equations are respectively

$$R_1(\xi, \eta, \xi_0, \eta_0) = F_3(m_2, m_4, 1 - m_2, 1 - m_4, 1, z_{16}, z_{18}), \tag{144}$$

$$R_2(\xi, \eta, \xi_0, \eta_0) = F_3(m_1, m_3, 1 - m_1, 1 - m_3, 1, z_{19}, z_{22}), \tag{145}$$

$$R_3(\xi, \eta, \xi_0, \eta_0) = {}_2F_1(n, 1 - n; 1; z_0), \tag{146}$$

where $z_{16}$ and $z_{22}$ were given by (126) and (127). Similarly, $z_0$, $z_{19}$, and $z_{22}$ were defined by (16), (131), and (132) respectively. Note also that the roles of $\xi$ and $\eta$ are interchanged, as required. Call $R_4(\xi, \eta, \xi_0 \eta_0)$ the Riemann function for (143), which was defined by Equations (86)–(88) (again interchanging the roles of $x$ and $y$ with $\xi$ and $\eta$ as required).

Start by combining (140) and (141), which produces the equation

$$U_{\xi\xi} - U_{\eta\eta} + \left[ \lambda_1^2 \left( \frac{m_2(1 - m_2)}{\sin^2 \lambda_1 \eta} + \frac{m_4(1 - m_4)}{\cos^2 \lambda_1 \eta} \right) - \lambda_2^2 \left( \frac{m_1(1 - m_1)}{\sinh^2 \lambda_2 \xi} - \frac{m_3(1 - m_3)}{\cosh^2 \lambda_2 \xi} \right) \right] U = 0. \tag{147}$$

The Riemann function for (147) is then

$$
\begin{aligned}
R(\xi, \eta, \xi_0, \eta_0) = \ & F_3(m_2, m_4, 1 - m_2, 1 - m_4, 1, z_{33}, z_{34}) \\
& + \int_{\xi - \xi_0}^{\eta - \eta_0} F_3(m_2, m_4, 1 - m_2, 1 - m_4, 1, u_2(t), u_3(t)) \times \\
& \quad \frac{\partial}{\partial t} F_3(m_1, m_3, 1 - m_1, 1 - m_3, 1, v_2(t), v_3(t)) \, dt,
\end{aligned} \tag{148}
$$

where

$$u_2(t) = \frac{\cos \lambda_1(\eta - \eta_0) - \cos \lambda_1 t}{2 \sin \lambda_1 \eta \sin \lambda_1 \eta_0}, \qquad u_3(t) = \frac{\cos \lambda_1(\eta - \eta_0) - \cos \lambda_1 t}{2 \cos \lambda_1 \eta \cos \lambda_1 \eta_0},$$

$$v_2(t) = \frac{\cosh \lambda_2 t - \cosh \lambda_2(\xi - \xi_0)}{2 \sinh \lambda_2 \xi \sinh \lambda_2 \xi_0}, \qquad v_3(t) = \frac{\cosh \lambda_2(\xi - \xi_0) - \cosh \lambda_2 t}{2 \cosh \lambda_2 \xi \cosh \lambda_2 \xi_0}.$$

The constants $\lambda_1$ and $\lambda_2$ were introduced trivially into (140) and (141). However, after applying these two equations to the addition formula, the ratio $\lambda_1/\lambda_2$ becomes essential. In other words, it is possible to transform away either $\lambda_1$ or $\lambda_2$, but it is no longer possible to eliminate both $\lambda_1$ and $\lambda_2$ from (147) via a change of variables. Effectively, a five parameter Riemann function has been obtained. It is useful to write the equation as (147) though, which at first glance incorporates six parameters, as the equation is then symmetric. This is now the most general self-adjoint equation for which the Riemann function is known. It incorporates one more essential parameter than Chaundy's Equation (17). It was first published in [68].

The next obvious choice is to combine Equations (142) with (140) and (141) to produce

$$U_{\xi\xi} - U_{\eta\eta} + \left[ \frac{m_1(1 - m_1)}{\xi^2} - \frac{m_2(1 - m_2)}{\sin^2 \eta} - \frac{m_4(1 - m_4)}{\cos^2 \eta} \right] U = 0, \tag{149}$$

$$U_{\xi\xi} - U_{\eta\eta} + \left[ \frac{m_1(1 - m_1)}{\xi^2} - \frac{m_2(1 - m_2)}{\sinh^2 \eta} + \frac{m_3(1 - m_3)}{\cosh^2 \eta} \right] U = 0. \tag{150}$$

The Riemann function for (150) was first given in [51] by using the addition formula. However, both (149) and (150) can be obtained as confluent reductions of (147). To see this, firstly let $m_3(1 - m_3) = 0$ in (147). Then if we take the limit as $\lambda_2 \to 0$, we obtain (149). In a completely analogous way, to recover (150), start with (147), set $m_4(1 - m_4) = 0$, and again take the limit as $\lambda_1 \to 0$. The result is (150) as desired.

In order to obtain truly new Riemann functions, Equations (140)–(142) need to be combined with (143). Hence, for the equations

$$U_{\xi\xi} \quad - \quad U_{\eta\eta} - \left[ \frac{m_2(1 - m_2)}{\sin^2 \eta} + \frac{m_4(1 - m_4)}{\cos^2 \eta} + (1 + \delta\xi) \right] U = 0, \tag{151}$$

$$U_{\xi\xi} \quad - \quad U_{\eta\eta} - \left[ \frac{m_2(1 - m_2)}{\sinh^2 \eta} - \frac{m_3(1 - m_3)}{\cosh^2 \eta} + (1 + \delta\xi) \right] U = 0, \tag{152}$$

$$U_{\xi\xi} \quad - \quad U_{\eta\eta} - \left[ \frac{m_1(1 - m_1)}{\eta^2} + (1 + \delta\xi) \right] U = 0, \tag{153}$$

the Riemann functions are respectively

$$R(\xi, \eta, \xi_0, \eta_0) = R_1(\xi, \eta, \xi_0, \eta_0) + \int_{\xi - \xi_0}^{\eta - \eta_0} R_1(t, \eta, 0, \eta_0) \frac{\partial}{\partial t} R_4(\xi, t, \xi_0, 0) dt,$$

$$R(\xi, \eta, \xi_0, \eta_0) = R_2(\xi, \eta, \xi_0, \eta_0) + \int_{\xi - \xi_0}^{\eta - \eta_0} R_2(t, \eta, 0, \eta_0) \frac{\partial}{\partial t} R_4(\xi, t, \xi_0, 0) dt,$$

$$R(\xi, \eta, \xi_0, \eta_0) = R_3(\xi, \eta, \xi_0, \eta_0) + \int_{\xi - \xi_0}^{\eta - \eta_0} R_3(t, \eta, 0, \eta_0) \frac{\partial}{\partial t} R_4(\xi, t, \xi_0, 0) dt,$$

where $R_1(\xi, \eta, \xi_0, \eta_0)$, $R_2(\xi, \eta, \xi_0, \eta_0)$, and $R_3(\xi, \eta, \xi_0, \eta_0)$ are given by (144)–(146). $R_4(\xi, \eta, \xi_0, \eta_0)$ is defined by (86)–(88) and the roles of $\xi$ and $\eta$ are interchanged as required. The final possibility is to use (143) twice. That is, to take the equation

$$U_{\xi\xi} - U_{\eta\eta} - (\delta\xi - \omega\eta)U = 0. \tag{154}$$

The Riemann function for (154) is given by

$$R(\xi, \eta, \xi_0, \eta_0) = R_4(\xi, \eta, \xi_0, \eta_0) + \int_{\xi - \xi_0}^{\eta - \eta_0} R_5(t, \eta, 0, \eta_0) \frac{\partial}{\partial t} R_4(\xi, t, \xi_0, 0) dt,$$

where $R_4(\xi, \eta, \xi_0, \eta_0)$ remains unchanged. $R_5$ is obtained by interchanging the roles of $\xi$ and $\eta$ in (86)–(88). The Riemann function for (154) first appeared in [69] for the case where $\delta = \omega$. The full equation has not been published before.

Equations (151) to (154) admit no point symmetries. Likewise, they have no non-trivial Lie-Bäcklund symmetries. Hence, they fall outside the equivalence classes of Riemann functions studied in Sections 4.1 and 4.5. In this way, (151)–(154) represent a completely new class of Riemann functions.

In principle, all the Equations (140)–(143) are applicable to the multiplication Formula (94). However, no closed form Riemann function has been found using (94). This is left as an open problem for the interested reader.

## 6. Conclusions

Extending the review undertaken by Copson in 1958, seven additional approaches to the original six documented methods, have been detailed. Listing them chronologically, they are:

- Lie point symmetries.
- The method of successive approximations.
- The addition formula.
- Laplace transforms.
- The multiplication formula.
- Finite groups.
- Lie-Bäcklund symmetries.

Furthermore, the equivalences generated by point symmetries, where the governing equation admits a symmetry algebra isomorphic to $SL(2,R)$, have been clarified. This has now been complemented by a new equivalence class of Riemann functions which is obtainable only from a generalized symmetry and admits no non-trivial point symmetries. Hence, we have obtained a useful new diagnostic tool in the hunt for equivalences amongst PDEs and their Riemann functions. By combining several of the solution techniques, several new Riemann functions were then derived that admitted no symmetries whatsoever.

To conclude, other papers have also been published since Copson's review. For instance, the idea of the complex Riemann function has been considered [38,72–74]. It is largely based on the work of Vekua [75]. Qualitative properties of the Riemann function have also been explored [76–78]. As far as other qualitative properties go, only the "duality" property is known [40]. Higher dimensional Riemann functions have also been explored by Zhegalov [79] and Koshcheeva [80].

**Funding:** This research was funded by an Australian Postgraduate Award.

**Acknowledgments:** The referees are thanked for their feedback. The author is also grateful to Edward D. Fackerell and Christopher M. Cosgrove for their guidance on this work.

**Conflicts of Interest:** The author declares no conflict of interest.

## Appendix A. Hypergeometric Functions

Here, the definitions and key properties of the hypergeometric functions seen in this paper are included for reference. Greater detail can be found in [33,36]. The functions are included in the order in which they appeared.

- $_2F_1(a,b;c;z)$

$$_2F_1(a,b;c;z) = \sum_{n=0}^{\infty} \frac{(a)_n(b)_n z^n}{(c)_n n!},$$

where

$$(a)_n = \frac{\Gamma(a+n)}{\Gamma(a)}, \tag{A1}$$

$$(a)_0 = 1, \quad (a)_n = a(a+1)\dots(a+n-1), \quad n = 1,2,3,\dots$$

This function is a particular solution of the equation

$$z(1-z)\frac{d^2U}{dz^2} + [c - (a+b+1)z]\frac{dU}{dz} - abU = 0.$$

Of particular note is the relationship between the Legendre function and the hypergeometric function

$$P_m(1-2z) = {}_2F_1(m, 1-m, 1, z).$$

- $\Xi_2(a,b,c,z_1,z_2)$

$$\Xi_2(a,b,c,z_1,z_2) = \sum \frac{(a)_m(b)_n}{(c)_{m+n} m! n!} z_1^m z_2^n,$$

where $(a)_m$ etc. are given by (A1). This function is the solution to the system of equations

$$z_1(1 - z_1)\frac{\partial^2 U}{\partial z_1^2} + z_2\frac{\partial^2 U}{\partial z_1 \partial z_2} + [c - (a + b + 1)z_1]\frac{\partial U}{\partial z_1} - abU = 0,$$

$$z_2\frac{\partial^2 U}{\partial z_2^2} + z_1\frac{\partial^2 U}{\partial z_1 \partial z_2} + c\frac{\partial U}{\partial z_2} - U = 0.$$

- $F_3(a, a', b, b', c, z_1, z_2)$

$$F_3(a, a', b, b', c, z_1, z_2) = \sum \frac{(a)_m (a')_n (b)_m (b')_n}{(c)_{m+n} m! n!} z_1^m z_2^n,$$

where $(a)_m$ etc. are given by (A1). This function is the solution to the system of equations

$$z_1(1 - z_1)\frac{\partial^2 U}{\partial z_1^2} + z_2\frac{\partial^2 U}{\partial z_1 \partial z_2} + [c - (a + b + 1)z_1]\frac{\partial U}{\partial z_1} - abU = 0,$$

$$z_2(1 - z_2)\frac{\partial^2 U}{\partial z_2^2} + z_1\frac{\partial^2 U}{\partial z_1 \partial z_2} + [c - (a' + b' + 1)z_2]\frac{\partial U}{\partial z_2} - a'b'U = 0.$$

- $F_B(a_1, \ldots, a_n, b_1, \ldots, b_n, c, z_1, \ldots, z_n)$

$$F_B(a_1, \ldots, a_n, b_1, \ldots, b_n, c, z_1, \ldots, z_n) =$$
$$\sum \frac{(a_1)_{m_1} \ldots (a_n)_{m_n} (b_1)_{m_1} \ldots (b_n)_{m_n}}{(c)_{m_1 + \ldots + m_n} m_1! \ldots m_n!} z_1^{m_1} \ldots z_n^{m_n}.$$

where $(a_1)_{m_1}$ etc. are given by (A1). This function satisfies the system of equations

$$z_j(1 - z_j)\frac{\partial^2 U}{\partial z_j^2} + \sum_{k \neq j} z_k\frac{\partial^2 U}{\partial z_j \partial z_k} + [c - (a_j + b_j + 1)z_j]\frac{\partial U}{\partial z_j} - a_j b_j U = 0,$$

for $i, j \leq n$.

- $J_n(z)$

$$J_n(z) = \sum_{r=0}^{\infty} \frac{(-1)^n z^{n+2r}}{2r!(n+r)!}.$$

This function is a particular solution of the equation

$$\frac{d^2 U}{dz^2} + \frac{1}{z}\frac{dU}{dz} + \left(1 - \frac{n^2}{z^2}\right)U = 0.$$

- $F_4(a, b, c, c', z_1, z_2)$

$$F_4(a, b, c, c', z_1, z_2) = \sum \frac{(a)_{m+n}(b)_{m+n}}{(c)_m (c')_n m! n!} z_1^m z_2^n,$$

where $(a)_{m+n}$ etc. are given by (A1). This function is the solution of the system of equations

$$z_1(1-z_1)\frac{\partial^2 U}{\partial z_1^2} - z_2^2\frac{\partial^2 U}{\partial z_2^2} \;-\; 2z_1z_2\frac{\partial^2 U}{\partial z_1 \partial z_2} + \left[c - (a+b+a)z_1\right]\frac{\partial U}{\partial z_1}$$

$$-(a+b+1)z_2\frac{\partial U}{\partial z_2} - abU = 0,$$

$$z_2(1-z_2)\frac{\partial^2 U}{\partial z_2^2} - z_1^2\frac{\partial^2 U}{\partial z_1^2} \;-\; 2z_1z_2\frac{\partial^2 U}{\partial z_1 \partial z_2} + \left[c - (a+b+a)z_2\right]\frac{\partial U}{\partial z_1}$$

$$-(a+b+1)z_1\frac{\partial U}{\partial z_1} - abU = 0.$$

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
