# Peer review of "On the Riemann Function"

_mathematics, doi:10.3390/math6120316_

Reviewer 1 Report

The paper reviews the existing methods for finding Riemann functions. The Riemann functions play important role in solving some boundary-value problems for partial differential equations. The topic should be of interest to researchers in various fields like partial differential equations, special functions, applications of Lie groups, numerical methods and so on.

The paper is well-structured and provides a historical perspective on the development in the field – starting from Riemann’s original work to the modern use of symmetry groups for constructing Riemann functions. The strengths and the weaknesses of the presented methods are discussed and some open problems and directions for future research are pointed out.

The author finds and explains some non-trivial relations and equivalences between known Riemann functions. Finally, applying the so called Olevskii's addition formula on some known cases, the author constructs some new Riemann functions for equations that admit no symmetries.

I found the paper well-written and interesting to read. There are few misprints and other issues. Therefore, I recommend publication after minor revisions.

Corrections and suggestions:

Page 1, line 8 from bottom: "...Rieman..." should be "...Riemann..."

Page 3, line 15: "...all the hypergeomtric..." should be "...all the hypergeometric..."

Page 5, line 13 from bottom: "3. Method’s Not..." should be "3. Methods Not..."

Page 9, line 9: "...(43) satisifes (42)..." should be "...(43) satisfies (42)..."

Page 13, line 14: "...Riemann [25] in section (1)..." perhaps should be "...Riemann [25] in section (2)..." or "...Riemann [25] in section (2) method 1..."

Page 18, line 20: "Taking advantage of (1)..." could be misleading as it looks like a reference to equation (1). Perhaps one may say "Taking advantage of the transformation 1..."

Author Response

All the suggested amendments and corrections have been made. I thank the referee for this careful reading (and I apologize for not proofreading the manuscript more closely myself). 

Reviewer 2 Report

The main theme of the work concerns the Riemann’s method.

The author is inspired by the work published in 1958 by E. T. Copson, entitled " On the Riemann-Green Function" which, according to the author, represents the first review of Riemann's method.

In the present work, the author, having examined the solution methods cited by Copson in his article, he integrates these methods with those already formulated at the time of publication of the article and which they were not aware of by Copson.

Moreover, to these methods, the author adds the examination of further methods that have been formulated since 1958 to today, and critically, he identifies those that are equivalent to each other.

In the end, the author, by combining several of the solution techniques, derives new functions of Riemann that have not admitted symmetries whatsoever.

As we have said, the work starts from the aforementioned Copson's paper of 1958, however, it must be emphasized that this work extends the starting paper by completing it in the parts that Copson himself had neglected to mention.

The investigation is further enriched and very well completed by the critical analysis of further methodologies that over the years have emerged from the now distant 1958.

At the end of the work, the author, in an innovative way, proposes the construction of new functions of Riemann that do not fall into any of the previous casuistry.

The work is well structured, clear and linear in its development.

The results are well presented and the English language is up to the work.

Overall, it is a good paper in line with the editorial guidelines of the journal that provides new contributions to the topic addressed and reveals some new problem that will surely be better defined in a subsequent work.

Author Response

I thank the referee for the feedback.